# Healthcare Professionals’ Documentation in Supported Accommodation for People with Profound Intellectual Disabilities: An Educational Intervention Study

**DOI:** 10.3390/healthcare12161606

**Published:** 2024-08-12

**Authors:** Kjellaug K. Myklebust, Julia Bogen Ramstad, Solveig Karin Bø Vatnar

**Affiliations:** 1Faculty of Health Sciences and Social Care, Molde University College, P.O. Box 2110, 6402 Molde, Norway; 2Centre for Research and Education in Forensic Psychiatry, Oslo University Hospital, P.O. Box 4956, 0424 Oslo, Norway

**Keywords:** educational intervention, electronic health record, intellectual disability, staff documentation, supported accommodation, scale for the evaluation of staff-patient interactions in progress notes

## Abstract

Good-quality relationships in which individuals with profound intellectual disabilities (intelligence quotient, IQ < 20–25) are recognized by healthcare professionals (HPs) are essential for the quality of healthcare and promoting autonomy. This study examines the impact of an educational intervention on documentation of the interplay between HP and individuals receiving services in supported accommodation in Norway. An educational intervention study was designed to encourage HPs to document their approaches and interplay. The Scale for the Evaluation of Staff-Patient Interactions in Progress Notes (SESPI) was applied to measure documentation before and after the intervention. Journal notes written over a three-month period before the intervention and a three-month period after the intervention were measured. Prior to the intervention, only 23.1% of the journal notes described the resident’s experiences, increasing by 5.4% (*p* = 0.041) post-intervention. Practical solutions to individual experiences increased from 0.9% to 8.5% (*p* < 0.001). The educational intervention demonstrated a significant increase in the documentation of residents’ experiences and the interplay between HPs and residents. Future research should explore the generalizability of these findings. Incomplete documentation of HPs’ relational work conceals important aspects of the healthcare provided, potentially resulting in confining autonomy and participation for individuals with intellectual disabilities.

## 1. Introduction

Research on services for individuals with intellectual disabilities has emphasized the importance that healthcare professionals (HPs) listen empathetically and understand these people’s experiences. This view is shared by both people with intellectual disabilities and HP [1,2,3]. Qualitative studies have noted that individuals with intellectual disabilities have greater opportunities for influencing their everyday lives when HPs are attuned to their experiences and meet them with empathy [4,5,6,7]. Profound intellectual disability is often defined as an intelligence quotient (IQ) below 20–25 points or a developmental age below 24 months. Persons with profound intellectual disabilities communicate, for instance, by body movements, vocalizations, or other subtle signals [8,9]. HPs often face difficulties in recognizing the emotions and responses of individuals with profound intellectual disabilities [3,10]. This may challenge documentation of the relational aspects of care. Consequently, subsequent-shift HPs may miss vital information about individual experiences and HPs approaches previously attempted in response to those experiences. This can lead to individuals with profound intellectual disabilities not being understood and not receiving the support required to participate in meaningful activities [11].

In this study, we examined journal notes in electronic health records (EHRs) at a residential facility for people with profound intellectual disabilities (‘residents’ below). We investigated the extent to which HP documented residents’ experiences and the degree of documented HP approaches to these situations. Documentation of the care provided is legally required and is considered important to ensure good quality of care and safety for people with intellectual disabilities [12,13]. However, researchers have described the documentation in supported accommodation as fragmented, lacking a person-centered focus, and giving limited aid to facilitate good quality of care for the residents [12]. Other researchers have warned about the power of documenting seemingly neutral observations of residents’ behavior and that these texts play a crucial role in constructing a biased ‘reality’ [14,15]. Moreover, interviews with HP revealed that they adjusted their documentation according to their service organization’s expectations, even though their own approaches would have been different [16]. Nonetheless, it has been reported that documentation of care, particularly in supported accommodations for people with intellectual disabilities, is an underexplored research topic [12].

A literature search indicated that documentation of the interplay between HP and individuals with intellectual disabilities is insufficient. For example, there was a lack of documentation of challenging behavior and how professionals best approached such situations to prevent escalation [17]. Talman et al. found that HP documentation of what and how people with severe intellectual disabilities attempted to communicate was scarce [18]. To our knowledge, no previous studies have examined EHR content regarding individuals with profound intellectual disabilities and how HP approached them.

There is a significant gap between the literature’s emphasis on the importance of establishing relationships and empathy [19,20] and the extent of research concerning the documentation of relational work between HP and individuals with intellectual disabilities. Accordingly, examining the documentation of residents’ experiences and HP approaches to those experiences in a supported accommodation for persons with profound intellectual disabilities can provide valuable knowledge.

### Theoretical Frame

This study is based on a relational understanding of intellectual disabilities, whereby challenges in interplay are linked to both the individual with intellectual disabilities and the HP. In the literature, the ability of HP to attune to the person with intellectual disabilities is described as vital for the quality of the interaction [10,11,21]. The concept of attunement is based on Daniel Stern’s work and is closely related to empathy [22]. Stern analyzed mother–infant interaction and found that they attuned to each other’s emotional expressions through gestures, eye contact, and sounds. For example, if a mother attuned to her restless infant, the infant would become calm. Attunement is essential in therapeutic contexts, where the therapist’s ability to attune to the client is a prerequisite for the client to experience empathy [22]. Attunement is also highly relevant for encounters with people with communication impairments, where the professional’s interpretation of the individual’s signals about their experiences and ability to attune to them is crucial for how the relationship is experienced [3,19].

The context for this study was a residential facility for people with profound intellectual disabilities, including individuals with extensive challenges related to verbal communication. The study measured the degree and quality of interplay described in EHR journal notes. A reliability-tested tool, the Scale for the Evaluation of Staff-Patient Interactions in Progress Notes (SESPI), was used [23]. The aim was to examine the impact of an educational intervention on journal documentation of interplay between residents and HP in a municipality-supported accommodation by investigating the following research questions:How frequent were the residents’ experiences described in EHR journal notes prior to the intervention?To what extent was interplay between residents and HP documented prior to the intervention?What was the impact of the educational intervention regarding documentation describing residents’ experiences and resident–HP interplay?

## 2. Materials and Methods

This is a quantitative educational intervention study. The second author scored journal notes retrieved from EHRs prior to and after an educational intervention in the form of a brief educational sequence, which has been used in several studies aimed at influencing healthcare personnel [24,25]. The intervention description was guided by the Template for Intervention Description and Replication [26].

### 2.1. Setting and Sample

The setting for this study was a 24-hour supported accommodation for people with profound intellectual disabilities in a Norwegian municipality. The data were obtained from EHR journal notes describing the residents’ daily care. The accommodation had six separate living units for the residents, and the study included journal notes for all six residents. The residents ranged in age from 24 to 48 years, and all had extensive service needs. A total of 18 HPs worked in shifts at the facility. Four staff members held bachelor’s degrees in healthcare or social fields (e.g., nursing or social work), which included training in managing challenging communication and, to some extent, empathetically addressing challenging behavior. Nine staff members were healthcare assistants, while the remaining five were unskilled workers. According to Norwegian laws, they were responsible for providing holistic care, with each person being met individually [27]. HPs ranged in age from 20 to 54 years, and they had 1–32 years of professional experience in services for individuals with intellectual disabilities. All HPs wrote daily EHR journal notes for the residents they engaged with.

### 2.2. The Intervention

One of the authors, who was also the leader of the supported accommodation, conducted an educational intervention aimed at increasing the documentation of the residents’ experiences and HP–resident interplay. The same educational session was conducted over two different days to allow all staff HPs to participate. The intervention followed a written procedure to ensure consistency and contained the following:Presenting the purpose of the study and formal documentation requirements.Emphasizing the importance of comprehending residents’ experiences, attuning to their emotions in daily practice, and documenting that practice. At this point, the researcher highlighted that the HP appeared to have extensive knowledge about how to meet each resident’s individual needs, but it was uncertain to what degree these approaches were reflected and documented in the notes.Providing selected journal notes from this accommodation to illustrate the differences between those that described HP–resident interplay and those that lacked such descriptions.Offering a brief explanation of how journal notes would be measured according to the SESPI. The rationale for introducing the tool to the HP was twofold: to ensure transparency in the study and to provide training on how to document HP–resident interplay. Ten journal notes from the accommodation were used for pedagogical purposes to enhance the intervention. These journal notes were selected to facilitate discussions between the researcher and the HP about what constitutes attuned interplay and how such interactions can be documented. Through this process, the HP noted that they had more extensive interplay with residents than was reflected in their documentation. The researcher acknowledged HP perspectives and highlighted the potential benefits of recording interplay. It was emphasized that the goal was to provide honest descriptions of the interplay, including unsuccessful instances. Such documentation could provide valuable insights for improving future interplay.

### 2.3. The SESPI Tool

SESPI was originally developed and validated in a Norwegian psychiatric inpatient context and has recently been applied in other European studies. SESPI is reliability tested; Cronbach’s alpha for the entire instrument was 0.977, indicating a very high internal score between raters. The ICC was 0.770. Each step of the SESPI was also tested, and the alpha and ICC values varied between 0.970 and 0.992 [23,28,29,30]. The SESPI tool assesses the degree of description of the resident’s experience in a journal note and measures the degree to which it contains a description of the HP’s approach to the resident’s experience (HP–resident interplay). There are four categories for the description of HP–resident interplay: (a) no/unclear description of approach; (b) approach described, but it is unclear how the resident experienced the approach; (c) approach oriented towards practical solutions and not towards experiences and feelings; (d) both the HP’s approach and the resident’s response to it are described clearly; that is, the described interaction can be assessed regarding attunement. For instance, the following journal note was scored in category (d) “Fred did not want to eat dinner and became more and more restless. He repeatedly showed the communication sign for the day center. Then, the HP showed Fred a picture of the HP who was responsible for following him to the day center the next day. He seemed satisfied with this and went to eat dinner”. Only journal notes scored in category (d) can be assessed concerning attunement in terms of degrees of Failed or Successful attunement. The journal note about Fred was scored as *Successful attunement*; the HP sensed Fred’s worries, attuned to him, and managed to meet him in a way that left Fred more satisfied.

### 2.4. Procedures

HPs documented the care they provided in journal notes, which were written in EHRs under the “Plan/Report” heading. Initially, all notes under this heading were included in the study. However, three subcategories (doctor, scanned documents, medical journal) were deemed irrelevant for documenting HPs daily work and were therefore excluded from the study. Additionally, a small number of journal notes were also excluded because they were written by healthcare professionals who were not employed in the supported accommodation and did not participate in the intervention. Except for these few exceptions, all notes written in the six residents’ EHRs during a three-month period before the intervention (17 December 2020 to 16 March 2021) and a three-month period straight after the intervention (24 March 2021 to 24 June 2021) were included.

### 2.5. Training in SESPI and Test Scoring

One of the researchers, who was also a leader for the supported accommodation, received training in SESPI. Prior to the training session, the SESPI tool and a brief guideline were sent to the researcher via email. The guideline included sample journal notes and instructions on how to code them. During the 45 min training session, the researcher was introduced to the SESPI tool and scored 15 journal notes under the guidance of one of SESPI’s creators. A test was conducted to ensure the tool was being applied correctly, followed by trial scoring of 36 random journal notes from the six residents’ EHRs. The researcher, who had received the training and was familiar with the supported accommodation context, along with one of SESPI’s creators, who was unfamiliar with the health service context, independently scored the 36 notes. The initial scores showed some differences due to the researcher’s knowledge of the residents. Therefore, extra care was taken to score only what was explicitly described in the journal notes. New trial scorings followed and showed consensus in the use of SESPI; no need to adjust the original SESPI manual was identified. The consensus in the trial scoring supported the use of SESPI as an adequate analysis tool to answer the study’s research questions. A tool’s reliability is not a fixed quality but depends on the raters’ training in using the tool [31]. SESPI has previously shown good results regarding reliability. The training session, trial scores of 36 notes, additional training, and finally, new trial scores were important to ensure the correct use of SESPI, thus ensuring reliability. After this training process, the researcher, familiar with the accommodation context, further conducted all the SESPI scoring.

### 2.6. Measures

The intervention was carried out on 17 March 2021 and 23 March 2021. All journal notes written in the six residents’ EHRs over the three-month period before the intervention (17 December 2020–16 March 2021) were scored in SESPI, provided that the notes met the previously described inclusion criteria and did not fulfill any of the exclusion criteria. This constituted the study’s baseline (*N* = 742). To assess whether the intervention resulted in a change in journal note content, the notes written in the first three months after the intervention (24 March 2021 to 24 June 2021) were scored in SESPI (*N* = 834). The inclusion and exclusion criteria were consistent. The two measurement periods (three plus three months) included a total of 1576 journal notes from the six residents’ EHRs.

In addition to scoring the journal notes in SESPI, the period number prior to (1) and after (2) the intervention, resident number (1–6), shift type (day, evening, night, and other), education, gender, and age of HP, an HP’s relation to the resident (primary contact, secondary contact, and other), and number of years of work experience in total and at the supported accommodation were added as control variables.

Period, resident number, shift type, education, gender, relation to the resident, and the SESPI scores considering the description of experiences (Yes/No) were added as nominal variables. The SESPI measured HP–resident interplay with ordinal-level variables, while journal note number, age, years of work experience, and years of work experience in the supported accommodation were continuous scales.

### 2.7. Data Analysis and Statistics

Statistical analyses were performed using SPSS v. 28. First, descriptive analyses were performed, followed by simple cross-tabulations to determine whether there were differences pre- and post-intervention. To investigate specifically whether the intervention changed the proportion of journal notes that described residents’ experiences, any group differences between the journal notes before and after the intervention were tested using cross-tabulations and chi-square tests. Identical tests were conducted to investigate whether the intervention changed the proportion of journal notes describing HP approaches to residents’ experiences (the HP–resident interplay). In both cross-tabulations, the periods were entered as dependent variables. In the first cross-tabulation, the described/undescribed experience was entered as an independent variable. In the second cross-tabulation, SESPI measurement for HP–resident interplay constituted the independent variable, with categories (four categories, a–d) including “No description of experience”, “No/unclear description of approach”, “Approach described, but unclear how the resident experienced it”, “Approach oriented towards practical solutions (but without describing an attunement to the resident’s emotions)”, and “Both HP approach and resident response are clearly described”. Furthermore, multivariate logistic regression was used to investigate whether pre- and post-intervention changes could be due to control variables (resident number (1–6), shift type (day, evening, night, and other), education, gender, age, relation to the resident (primary contact, secondary contact, team leader, and other), total years of work experience, and years of work experience in supported accommodation). Univariate regression analyses were performed individually for each control variable. Only the control variables that were significant (*p* < 0.05) in the univariate analyses were included in the multivariate regression models. All authors were involved in the statistical analysis.

## 3. Results

### 3.1. Documentation of Resident Experiences before the Intervention

During the assessment period before the intervention (baseline), a total of 742 EHR journal notes were examined. In 21.3% of these journal notes, we found descriptions of the resident’s experiences; 78.7% lacked such descriptions.

### 3.2. Documentation of Interplay between Residents and HP before the Intervention

Of the 742 journal notes that constituted the baseline, 17.4% described experiences without mentioning any approach to those experiences. In very few (0.9%) journal notes, both resident experiences and HP approaches to those experiences were described, but not how the resident had responded to the approach. The same amount (0.9%) described HP approaches oriented towards practical solutions but without describing an attunement to residents’ emotions. In 2.0% of the journal notes, we found descriptions of HP approaches to residents’ experiences and how the resident had responded, enabling an assessment of the quality of the interaction described in terms of successful or failed attunement. As noted, 78.7% of journal notes had no description of experiences and thus no description of approaches to experiences (see Table 1).

### 3.3. Documentation Prior to and after the Intervention

The proportion of journal notes describing resident experiences increased from 21.3% prior to the intervention to 26.7% after the intervention: χ^2^ (2) = 7.401, *p* = 0.025 (see Table 1). Analyses of descriptions of resident–HP interplay showed that the proportion of journal notes containing descriptions of practical approaches by HP increased from 0.9% to 8.5% after the intervention: χ^2^ (4) = 54.287, *p* < 0.001. Similarly, there was a decrease in the category “No/unclear description of approach” from 17.4% before to 14.7% after the intervention. The other categories remained stable (see Table 1).

Table 2 displays a logistic regression analysis using a multivariate model to assess whether the observed increase in the proportion of journal notes describing residents’ experiences remained significant when controlling for other relevant variables. The odds ratio shows a significant increase in the proportion of journal notes containing descriptions of resident experiences after the intervention when controlling for shift type, education level, and age of the journal note writer.

Table 3 shows an increase in the proportion of journal notes containing descriptions of interplay after the intervention. This growth consisted mainly of an increase in the proportion of journal notes that described approaches oriented towards practical solutions (but without describing attunement). This corresponds to the descriptive findings. The increase in descriptions of practical approaches persisted as a significant change when controlling for the model’s explanatory variables (shift type, age, and education level). Additionally, the analyses showed a statistical trend in growth in the proportion of journal notes describing interplay clearly enough to be assessed in terms of attunement, but this change was not significant (*p* = 0.064, see Table 3). The descriptive analyses also suggested an increasing trend towards full interaction descriptions after the intervention (see Table 1).

## 4. Discussion

The findings suggest that the educational intervention had a positive impact on the frequency and level of documenting residents’ experiences and interplay. There was a significant increase in the documentation of those experiences and practical approaches from HP related to them. Although there was a rise in the proportion of journal notes that described HP–resident interplay with enough clarity to be evaluated for attunement, that increase was not statistically significant. Previous studies have evaluated staff training interventions for enhancing sensitive approaches when interacting with individuals with profound intellectual disabilities [32]. However, to our knowledge, this is the first to evaluate an intervention designed to improve the documentation of these aspects.

### 4.1. Documentation of Experiences

Documentation of resident experience improved after the intervention; however, only about a quarter of journal notes described the resident’s experience. Our findings align with a quantitative study on individuals with profound intellectual disabilities that reported a lack of documentation of individuals’ experiences and contextual information while documenting challenging behaviors [17]. Previous research has shown that staff struggled to comprehend the non-verbal communication and emotions of individuals with profound intellectual disabilities [2,13]. This may explain the low proportion of journal notes containing descriptions of resident experiences. However, this aspect requires further investigation.

### 4.2. Documentation of HP–Resident Interplay

The main difference between journal notes before and after the intervention was the proportion of notes that described approaches oriented towards practical solutions. This category increased significantly. A typical journal note in this category was “Liza cried and was clearly uncomfortable. Administered pain reliever medication”. The intervention aimed to encourage HP to document their approaches to residents’ experiences. The increase in practical approach descriptions and the decline in notes lacking any approach to the resident’s experience are positive developments. However, the interplay still lacked sufficient description to evaluate it in terms of successful or failed attunement. After the intervention, there was an increase in such descriptions, although it was not statistically significant. Nevertheless, this finding may suggest the potential for changes in journal documentation. The journal note above could be evaluated for attunement if it included approaches like comforting Liza or exploring the reasons for her crying, rather than just administering medication. Qualitative researchers have advocated a shift in documentation practices that go beyond focusing solely on individuals with intellectual disabilities and taking into account their interplay with HP [14,15]. A systematic review of intervention studies intended to alter staff communication in services with individuals with profound intellectual disabilities found that only 6 of 11 studies showed statistically significant changes after that intervention [32]. In comparison, the present study managed to demonstrate a statistically significant improvement in the documentation of interplay, suggesting the potential for further developing this educational intervention that promotes documentation of HP–resident interplay.

### 4.3. Clinical Implications

Good-quality relationships with HP attuned to individuals’ experiences are core elements in caring for people with profound intellectual disabilities [11,21]. This study found limited documentation of these aspects, but a simple intervention increased the proportion of journal notes, including resident experiences and how HP approached them. If supported by other studies, such interventions might provide empirical knowledge to develop improved documentation practices in the care of people with intellectual disabilities. Improving staff communication with individuals with limited verbal communication capability has been a goal in several recent studies [32,33]. Documentation of HP–resident interplay may play a vital role in enhancing HP communication tailored to an individual’s needs [15,16]. For instance, HP can gain insights into approaches that enhance an individual’s participation by reading about their colleagues’ successes and failures in meeting that person’s emotional and practical needs during previous shifts. Adequate documentation from one shift to the next is crucial in facilitating safe, continuous, and good-quality care for people with intellectual disabilities living in supported accommodation [12]. Incomplete documentation of relational work by HP renders this crucial aspect invisible, possibly leading to an understanding of services for people with intellectual disabilities where relational work takes a back seat to one-sided task-oriented and medical work, which could limit the possibilities for autonomy and participation for people with intellectual disabilities [6].

### 4.4. Strengths and Limitations

The study has provided a comprehensive overview of reporting practices before and after intervention, thus contributing to empirical knowledge regarding documentation of the interplay between individuals with profound disabilities and their HP. The use of a recommended template for intervention studies [26], a reliability-tested tool to measure documented interplay [23], and regression analyses with multiple control variables to address potential confounders strengthened the study’s internal validity. The study’s design was not intended to investigate causal relationships, but the regression analyses confirmed a correlation between the intervention and the observed changes in the journal notes. This indicates that the intervention likely contributed to that change. The study used a substantial number of journal notes from all EHRs in the supported accommodation, which suggests that the findings are representative of (at least) that accommodation. However, it had only six residents and a comparatively modest number of EHRs. A small sample size is described as a common limitation in intervention studies conducted in similar contexts [32]. Comparing this study’s results with other studies has been challenging due to the lack of similar studies with comparable contexts and samples. Further research is needed to gauge the generalizability of these findings.

The leader of the supported accommodation conducted the intervention. Being familiar with the HP context is crucial for impactful educational interventions [25]. Using real journal notes as examples during the intervention may have helped clarify the study’s purpose. However, conducting the intervention as a leader may have deterred some employees from asking critical questions, potentially weakening the intervention. Although simple and brief, the intervention had a positive impact on documentation practices, but the sustainability of its effects beyond the three-month follow-up is unclear.

SESPI is a reliability-tested tool applied in mental healthcare settings [23,28,29,30]. During the initial scoring process, we found no need to adjust the SESPI manual. However, we faced some challenges in scoring certain journal notes during the study. For instance, unconventional expressions of pain by individuals with intellectual disabilities made pain assessment difficult when the journal note described the resident laughing despite being in a possibly painful situation [34]. The manual lacked examples of atypical pain expressions, but the researchers discussed them and reached consensus on the relevant categories. Additional context-specific examples should be included in the SESPI manual for future studies.

The measured units in the current study were journal notes written by HP; correspondingly, a significant limitation was the absence of the residents’ own perspectives on the content. However, the residents in the present study had extensive challenges related to verbal communication, implying limitations regarding participating in written documentation. Collaboration with people with intellectual disabilities in care planning and documentation of care is scarce but called for [35]. Developing methods for including residents’ perspectives when assessing documentation is pivotal for future research.

## 5. Conclusions

After the educational intervention, there was an increase in the proportion of journal notes describing residents’ experiences and practical approaches, indicating a change in the HPs’ process of writing journal notes. Thus, this educational intervention study is an important first step towards developing a documentation practice containing descriptions of residents’ experiences and the interplay between residents and HP. Improving documentation in supported accommodation is considered crucial for improving the quality of healthcare. People with profound intellectual disabilities have individual ways of communicating (e.g., sounds and body language). Enhancing the documentation of their experiences can contribute to improved care, leading to a better quality of life and opportunities to live like others do. To enhance generalizability, future studies should include more residents in multiple supported accommodations when repeating the intervention.

## Figures and Tables

**Table 1 healthcare-12-01606-t001:** Documentation of residents’ experiences and interplay between residents and healthcare professionals (HPs) before and after intervention: descriptive statistics.

		Before Intervention, T0 (*n* = 742) % (*n*)	After Intervention, T1 (*n* = 834) % (*n*)	Total (*N* = 1576) % (*n*)	*p*-Value
Does the journal note describe the resident’s experience?	No	78.7 (584)	73.3 (611)	75.8 (1195)	0.025
	Yes	21.3 (158)	26.7 (223)	24.2 (381)	
Description of HP approach and resident response	No description of the resident’s experience	78.7 (584)	73.3 (611)	75.8 (1195)	≤0.001
	Description of the resident’s experience, but there is no/unclear description of approach	17.4 (129)	14.7 (123)	16.0 (252)	
	Approach described, but it is unclear how the resident experienced it	0.9 (7)	0.2 (2)	0.6 (9)	
	Approach oriented toward practical solutions and not toward experiences and feelings	0.9 (7)	8.5 (71)	4.9 (78)	
	Both HP approach and resident response are clearly described	2.0 (15)	3.2 (27)	2.7 (42)	

**Table 2 healthcare-12-01606-t002:** Journal notes’ descriptions of resident experiences before intervention (baseline) compared to after intervention (*N* = 1576).

Independent Variable	Adjusted Odds Ratio (OR)	(95% CI)	*p*-Value
**Description of experience (dichotomous)**	1.295	1.011–1.659	0.041
**Shift type**			<0.001
Day			(ref.)
Evening	1.149	0.920–1.435	0.220
Night	0.320	0.208–0.490	<0.001
Other	0.718	0.043–12.016	0.818
**Education**			0.011
Unskilled			(ref.)
Healthcare worker without bachelor degree	0.717	0.500–1.028	0.071
Bachelor in Social Education	0.215	0.078–0.594	0.003
Bachelor in Nursing	0.190	0.020–1.772	0.145
Bachelor in Child Welfare	0.596	0.379–0.938	0.025
Bachelor in Social Work	0.563	0.357–0.888	0.013
**Age**	0.974	0.962–0.987	<0.001

**Table 3 healthcare-12-01606-t003:** Journal notes’ descriptions of resident–healthcare professional (HP) interplay before intervention (baseline) and after intervention (*N* = 1576).

Independent Variable	Adjusted Odds Ratio (OR)	(95% CI)	*p*-Value
**Degree of described resident–HP interplay**			<0.001
No description of resident experience			(ref.)
Description of resident experience, but no/unclear description of approach	0.894	0.673–1.188	0.440
Approach described, but unclear how the resident experienced it	0.379	0.075–1.910	0.240
Approach oriented towards practical solutions and not towards experiences and feelings	9.766	4.388–21.737	<0.001
Both HP approach and resident response are clearly described	1.857	0.964–3.577	0.064
**Shift type**			<0.001
Day			(ref.)
Evening	1.186	0.948–1.485	0.135
Night	0.430	0.312–0.592	<0.001
Other	0.307	0.015–6.436	0.447
**Age**	0.972	0.961–0.984	<0.001
**Level of education**	0.753	0.587–0.965	0.025

Note: The level of education variable has been dichotomized into having or not having a bachelor’s degree.

## Data Availability

The datasets generated and analyzed during the current study are not publicly available due to the conditions for ethical approval but are available from the corresponding author on reasonable request.

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
