# Peer review of "Healthcare Professionals’ Documentation in Supported Accommodation for People with Profound Intellectual Disabilities: An Educational Intervention Study"

_healthcare, 2024, doi:10.3390/healthcare12161606_

Round 1

Reviewer 1 Report

Comments and Suggestions for Authors

Congratulations to the authors, the study addresses an important issue in the care of individuals with profound intellectual disabilities. With the suggested improvements, the manuscript will make a valuable contribution to the field.

Introduction: The introduction offers a general overview of the study's context and significance. However, it could benefit from a more detailed background on the challenges faced in documenting the care of individuals with profound intellectual disabilities. Including additional references to recent studies would provide a more robust foundation for the research context. Consider elaborating on the specific challenges healthcare professionals face in documenting the experiences of individuals with profound intellectual disabilities. This would provide a clearer rationale for the study.

Research Design: The pre-post intervention study design is appropriate for the research questions. The use of SESPI for evaluating documentation is suitable and well-justified. However, more details on control measures and how potential confounders were addressed would strengthen the validity of the design.

Methods: While the methods section provides a basic outline, it lacks detailed descriptions of the intervention procedures and the criteria for selecting journal notes. More comprehensive information on the training provided to HP and the scoring process for SESPI would enhance replicability. Provide a more detailed description of the educational intervention, including specific activities and materials used. Clarify the criteria for journal note selection and the training process for SESPI scoring.

Results: The results are presented using tables and statistical analysis. 

Conclusions: The conclusions are well-supported by the data presented. The discussion effectively ties the findings to the research questions and highlights the practical implications for improving documentation practices. Expand on clinical practice and policy implications, particularly how improved documentation can affect patient outcomes.

Author Response

Comment 1

Congratulations to the authors, the study addresses an important issue in the care of individuals with profound intellectual disabilities. With the suggested improvements, the manuscript will make a valuable contribution to the field. Response 1: Thank you for your positive feedback and suggestions. We have carefully considered and incorporated your suggestions into our revisions and believe the manuscript is now improved and strengthened.

Comment 2 

Introduction: The introduction offers a general overview of the study's context and significance. However, it could benefit from a more detailed background on the challenges faced in documenting the care of individuals with profound intellectual disabilities. Including additional references to recent studies would provide a more robust foundation for the research context. Consider elaborating on the specific challenges healthcare professionals face in documenting the experiences of individuals with profound intellectual disabilities. This would provide a clearer rationale for the study.

Response 2Thank you for an important reminder. We have added two references covering the challenges of communicating with people with profound intellectual disabilities (reference numbers 8 and 9) and described that this might challenge documentation of the relational aspects of care. Please see lines 38- 41, page 1. Documentation of care in this context is generally an unexplored research topic (This is described in lines 60-61, page 2).

Comment 3

Research Design: The pre-post intervention study design is appropriate for the research questions. The use of SESPI for evaluating documentation is suitable and well-justified. However, more details on control measures and how potential confounders were addressed would strengthen the validity of the design. Response 3: Thank you for pointing out that this is unclear. In addition to the text already covering the issue (see descriptions of control variables lines 215-224, page 5, and how we investigated pre- and post-intervention changes related to control variables lines 241-248), we have extended this issue in the section Strength and limitations, see changes in lines 366-367, page 10.

Comment 4

Methods: While the methods section provides a basic outline, it lacks detailed descriptions of the intervention procedures and the criteria for selecting journal notes. More comprehensive information on the training provided to HP and the scoring process for SESPI would enhance replicability. Provide a more detailed description of the educational intervention, including specific activities and materials used. Response 4a: Thank you, we agree with this comment. We have clarified that it was the same educational intervention that was held on two days (Lines 128-129, page 3). We have also revised The Intervention section: We have described the educational intervention in more detail. And we have explained the rationale for presenting the SESPI to the staff. See lines 133-136 and 141-144, page 3. Clarify the criteria for journal note selection and the training process for SESPI scoring.  Response 4b: Thank you. We have revised the Procedures section. With this, we have clarified that this study includes and measures all notes written in the EHR in the given time periods for the six residents (given that the notes fulfill the inclusion criteria and not the exclusion criteria). See lines 174-183, page 4. We also agree that the training process for SESPI scoring needed clarification, and we have added detailed information about the training session. Please see lines 186-196 of page 4.

Comment 5 Results: The results are presented using tables and statistical analysis. 

Comment 6

Conclusions: The conclusions are well-supported by the data presented. The discussion effectively ties the findings to the research questions and highlights the practical implications for improving documentation practices. Expand on clinical practice and policy implications, particularly how improved documentation can affect patient outcomes. Response to comment 6: Thank you. We agree and have added a couple of sentences about the importance of documenting the experiences, the challenges when the person mostly uses non-verbal language, and the fact that documenting the experiences may lead to a better quality of life and increased autonomy. See lines 408-411, page 11.

.

Reviewer 2 Report

Comments and Suggestions for Authors

Thank you for submitting this interesting article. It is very important for all people interacting witha person with intellectual disability to consider them hollistically. Notes are very important in this regard.

My comments are as follows

1. Please include definition of PROFOUND intellectual disability. Profound used in abstract and text. 

2. Please provide information on Healthcare Professionals qualifications,responsibilities etc.  What level of training have they with respect to challenging behaviours?  is there a standard qualification in Norway?

3. Terms used in Norway may not be applicable elsewhere.

4. Abstract should indicate Norway.. people often scan Abstracts to see if any applicable to their own setting.

I agree this is an important first step.

Well done.

Author Response

Thank you for submitting this interesting article. It is very important for all people interacting witha person with intellectual disability to consider them hollistically. Notes are very important in this regard. Thank you for your positive feedback on our manuscript.

My comments are as follows

Comment 1: Please include definition of PROFOUND intellectual disability. Profound used in abstract and text.

Response 1: Thank you. We agree that it is important to provide a clear definition of "profound intellectual disability" in both the main text and the abstract. We have made the necessary revisions and included a definition with an explanation in the main text (lines 38-41, page 1) and a concise definition in the abstract (line 13, page 1). We included two new literature references on this matter (reference numbers 8 and 9). Thank you for helping us improve the clarity of our manuscript.

Comment 2a: Please provide information on Healthcare Professionals qualifications,responsibilities etc.  What level of training have they with respect to challenging behaviours?

Response 2a: Thank you for pointing this out. We have considered your feedback and have, in addition to the information in Table 2, added more information about the healthcare professionals' qualifications, responsibilities, and training regarding challenging behavior, see lines 116-121, page 3. We also included a literature reference to Norwegian laws, see reference number 27.

Comment 2b: is there a standard qualification in Norway? Response 2b: No, there is no standard qualification in Norway. It is quite common for staff members to come from a heterogeneous group, with some holding bachelor's degrees in nursing or social fields, while others may be unskilled workers.

Comment 3: Terms used in Norway may not be applicable elsewhere.

Response 3: We agree and have elaborated on the bachelor degrees in lines 116-121, page 3. With this we think our manuscript uses applicable terms.

Comment 4: Abstract should indicate Norway.. people often scan Abstracts to see if any applicable to their own setting.

Response 4: Agree; we have accordingly added this to the abstract (line 16, page 1).

I agree this is an important first step. Thank you.

 Well done. Thank you!

Reviewer 3 Report

Comments and Suggestions for Authors

this is an interesting paper showing the significant effect of training on documentation of health care. I think there are a number of important issues that must be addressed:

Title: Since this is an experimental study, it is better to express the title so that the outcome of interest will be evident.

Abstract: I think it is necessary to define the method of the study somehow in more detail. For instance, the time period following the intervention for measuring the outcome of the study is an important issue.

Keywords: OK; however, the used phrases are a bit long.

Introduction: OK.

Material and method: Please notice to the following issues in this section:

-          Please define the validity and reliability of the used measuring tool in more detail. It is especially necessary to show the reliability measure in your own study.

-          Please define how the proposed outcome was measured and who was involved in the measurement.

-          It is very important to address the time following the intervention after which the outcomes were measured in the study setting.

-          Since this is a before-after experimental study, it is not clear if the distinguished authors have used paired statistical methods.

-          The variables other than the outcomes (mentioned as control variables in the context of the paper) are not defined (especially in terms of their measurement).

According to the above-mentioned issues the following sections of results and discussion are not justifiable.

Comments on the Quality of English Language

I think the paper needs a number of minor revisions for English language. 

Author Response

Comment 1: this is an interesting paper showing the significant effect of training on documentation of health care. I think there are a number of important issues that must be addressed:

Response 1: Thank you for your valuable feedback and suggestions, we have carefully reviewed the issues you raised.

Comment 2: Title: Since this is an experimental study, it is better to express the title so that the outcome of interest will be evident.

Response 2: We appreciate your suggestion on how we can improve the title of our study. However, considering the absence of a control group, our study was not designed to measure a causal relationship between the intervention and the changes observed in the journal notes. Instead, we focused on examining and measuring the observed changes in the journal notes before and after the intervention. We measured a correlation between the intervention and changes in the journal notes after the intervention. We understand your concern regarding expressing the outcome in the title. We think that it may not be feasible to convey such complexity (the findings of correlation) in the title itself. However, we have addressed these points within the manuscript to provide an accurate and comprehensive understanding of our study's findings. We have made no changes to the Title, and frame the study as an educational intervention study.

Comment 3: Abstract: I think it is necessary to define the method of the study somehow in more detail. For instance, the time period following the intervention for measuring the outcome of the study is an important issue.

Response 3: Thank you, we agree and have added a sentence on lines 19-20, page 1, to provide more detail on the time periods measured. However, the abstract is limited to 200 words, making it difficult to elaborate further on the method. We hope the added sentence provides some clarity on this issue.

Comment 4: Keywords: OK; however, the used phrases are a bit long. Introduction: OK.

Material and method: Please notice to the following issues in this section:

Comment 5: Please define the validity and reliability of the used measuring tool in more detail. It is especially necessary to show the reliability measure in your own study.

Response to comment 5: Thank you for a helpful suggestion that we agree would improve the manuscript. We have elaborated on this issue and added Cronbach’s alpha and ICC values from previous studies when SESPI was reliability tested, please see lines 154-157, page 4. Reliability is strongly dependent on the correct use of the tool. Therefore, we have added more detailed information about the training and trial scorings to ensure reliability in our study. We also added a new literature reference on this matter (reference number 31). See lines 186-196, page 4, and 199-203 on pages 4-5.

Comment 6: Please define how the proposed outcome was measured and who was involved in the measurement.

Response to comment 6: The proposed outcome was measured by the validated instrument SESPI. For detailed information about how and who was involved, see chapters 2.5 and 2.6 (pages 4-5), and analysis and statistics are described in 2.7, page 5. The researcher who received extensive training in SESPI scored all journal notes. We clarified this in lines 203-204, page 5. Additionally we have added that all authors were involved in the statistical analysis, see line 248, page 5.

Comment 7: It is very important to address the time following the intervention after which the outcomes were measured in the study setting.

Response to comment 7: We totally agree and have tried to cover this issue in section 2.6. We have added a clarification about the measurement periods in lines 213-214, page 5, where we carefully describe the time periods covered by the current study.

Comment 8: Since this is a before-after experimental study, it is not clear if the distinguished authors have used paired statistical methods.

Response to comment 8: The current study is an educational intervention study, which means that the data material is not pared data. We tested possible differences in journal notes (N=1576) before (n=742) and after (n=834) the intervention with cross-tabulations and chi-square tests. See descriptions in the first paragraph of 2.6 Measures and in chapter 2.7 Data Analysis and Statistics. The journal notes were not pared, and multivariate logistic regression was used to adjust for confounders or other group differences between the measurements (See description in Chapter 2.7). We have not made any changes in the manuscript regarding this comment.

Comment 9: The variables other than the outcomes (mentioned as control variables in the context of the paper) are not defined (especially in terms of their measurement).

Response to comment 9: Thank you for allowing us to extend on this. Table 2 provides descriptions of experiences before and after the intervention. The period (3 months before and 3 months after the intervention) is described in lines 206-214, page 5. We have added a clarification of this in line 213, page 5. Descriptions of experiences were added as nominal variables; this is described in lines 220-222, page 5. All variables included in the analyses are presented in Table 2.

Table 3 provides descriptions of the resident-staff interplay before and after the intervention. The descriptions of how resident-staff interplay was measured is described in lines 222-224, page 5 and 237-241, page 5. We have added a clarification, line 237, page 5. See also Table 3.

Comment 10: According to the above-mentioned issues the following sections of results and discussion are not justifiable. Response to comment 10: Regarding this comment, we hope all the previous changes and clarifications have provided sufficient information.

Round 2

Reviewer 3 Report

Comments and Suggestions for Authors

thanks to distinguished authors, it seems al of the comments are addressed.